# LncRNA-SNPs in a Brazilian Breast Cancer Cohort: A Case-Control Study

**DOI:** 10.3390/genes14050971

**Published:** 2023-04-25

**Authors:** Carolina Mathias, Anelis Maria Marin, Ana Flávia Kohler, Heloisa Bruna Soligo Sanchuki, Natalie Sukow, Marcia Holsbach Beltrame, Suelen Cristina Soares Baal, Ana Paula Martins Sebastião, Enilze Maria de Souza Fonseca Ribeiro, Daniela Fiori Gradia, Mateus Nóbrega Aoki, Jaqueline Carvalho de Oliveira

**Affiliations:** 1Department of Genetics, Federal University of Parana, Graduate Program in Genetics, Curitiba 81310-020, Brazil; 2Laboratory of Applied Science and Technology in Health, Carlos Chagas Institute, Oswaldo Cruz Foundation (Fiocruz), Curitiba 81310-020, Brazil; 3Department of Pathology, Hospital de Clínicas, Federal University of Paraná, Curitiba 81531-980, Brazil

**Keywords:** rs3803662, rs4415084, rs4784227, rs7716600, breast cancer susceptibility

## Abstract

Long noncoding RNAs (lncRNAs) are a class of non-coding RNAs that contain more than 200 nucleotides and exhibit a versatile regulatory capacity. Genomic alterations in lncRNAs have already been investigated in several complex diseases, including breast cancer (BC). BC is a highly heterogeneous disease and is the most prevalent cancer type among women worldwide. Single nucleotide polymorphisms (SNPs) in lncRNA regions appear to have an important role in BC susceptibility; however, little is known about lncRNA-SNPs in the Brazilian population. This study used Brazilian tumor samples to identify lncRNA-SNPs with a biological role in BC development. We applied a bioinformatic approach intersecting lncRNAs that are differentially expressed in BC tumor samples using The Cancer Genome Atlas (TCGA) cohort data and looked for lncRNAs with SNPs associated with BC in the Genome Wide Association Studies (GWAS) catalog. We highlight four lncRNA-SNPs—rs3803662, rs4415084, rs4784227, and rs7716600—which were genotyped in Brazilian BC samples in a case-control study. The SNPs rs4415084 and rs7716600 were associated with BC development at higher risk. These SNPs were also associated with progesterone status and lymph node status, respectively. The rs3803662/rs4784227 haplotype GT was associated with BC risk. These genomic alterations were also evaluated in light of the lncRNA’s secondary structure and gain/loss of miRNA binding sites to better understand its biological functions. We emphasize that our bioinformatics approach could find lncRNA-SNPs with a potential biological role in BC development and that lncRNA-SNPs should be more deeply investigated in a highly heterogeneous disease population.

## 1. Introduction

Breast cancer (BC) is the most common cancer among women worldwide, representing almost 26% of the total number of new cases of cancer diagnosed in 2020 [1]. Several risk factors have been associated with disease development, including environmental factors such as a sedentary lifestyle and alcohol consumption and intrinsic factors related to genetics [2].

Genetic susceptibility to BC can be related to germline mutations with high penetrance, such as those of the *BRCA1* and *BRCA2* genes, and due to the presence of single nucleotide polymorphisms (SNPs) in coding and non-coding regions [3].

Incorporating SNPs into risk prediction models, in combination with classical risk factors, can contribute to improving risk-based BC screening. Population-based screening programs aim to detect the disease at an early stage since effective treatment at this stage can lead to improved disease outcomes and lower mortality rates [4].

To identify genomic regions associated with BC, genome-wide association studies (GWAS) have already identified approximately 170 loci associated with BC risk, with the vast majority of GWAS identifying SNPs located in noncoding regions [5]. Among them, several long noncoding RNA (lncRNA) SNPs have been associated with BC, although this area remains relatively unexplored. lncRNAs are a class of noncoding RNAs that have more than 200 nucleotides with a versatile regulatory ability [6]. lncRNAs have been described as participating in breast cancer hallmarks, such as inducing proliferation, repressing apoptosis, and contributing to subtype differentiation [7,8].

In order to deepen the understanding of lncRNA-SNPs associated with BC risk, we conducted a GWAS data mining analysis to select some potential SNP candidates and then evaluated the selected lncRNA-SNPs in a Brazilian BC cohort.

## 2. Methods

### 2.1. GWAS Data Mining and lncRNAs Selection

As the first step, we performed a differential expression analysis using RNA-Seq HTSeq-FPKM from the TCGA Breast Cohort using as cut-offs fold change |>1.5| and *p* < 0.001, and then identified the ones that were classified as lncRNA. This analysis was made by comparing the tumor sample and its normal counterpart, and data were downloaded from XenaBrowser (https://xenabrowser.net/, accessed on 11 November 2022).

After the selection of differentially expressed (DE) lncRNA, we retrieved data from the GWAS Catalog (https://www.ebi.ac.uk/gwas/, accessed on 11 November 2022) using “breast carcinoma” as the keyword and downloaded the SNPs associated with each report [5]. Based on genomic coordinates, we mapped regions that contained lncRNAs. We selected the lncRNAs that both contain an associated SNP and were differentially expressed in breast tumors relative to normal tissue. After this filtering based on differentially expressed lncRNAs, we selected lncRNA-SNPs that exhibited minor allele frequencies higher than 0.20 in order to exclude rare alleles.

### 2.2. Study Cohort

The case-control analyses were performed using tumor samples of 291 patients with sporadic breast cancer (including all immunohistochemical subtypes) from the Hospital Nossa Senhora das Graças (HNSG), located in Curitiba, Paraná state, in southern Brazil. As a control group, we used peripheral blood samples of 370 women who were recruited on a voluntary basis and who declared they had no personal or family history of cancer with a mean age of 47.66 ± 4.69 years.

Patients and control samples were from the same region in the south of Brazil, most living in the metropolitan region of Curitiba. In the control group, ancestry information was obtained from self-reported patients’ records, with 84.7% white, 10.7% black or mixed-race, and 1.9% others. A subset of patients (*n* = 15) was genotyped using an SNP chip Illumina Infinium QC Array (Illumina Inc., San Diego, CA, USA), which contains 15,949 markers, including ~3000 ancestry informative markers (AIMs). Based on these ancestry data [9], the patients of this study clustered with the European-defined group from the 1000 Genome Project and with the Admixed Americans main group, mainly composed of Colombians and Mexicans, highlighting the marked ancestral heterogeneity of the Brazilian population [9].

The samples were collected with the approval of the Human Research Ethics Committee of the Health Sciences Sector of UFPR (CAAE: 67029617.4.0000.0102). All samplings and experiments followed relevant guidelines, Brazilian regulations, and ethical principles for human research in the Declaration of Helsinki. The project was described to all participants, and a written informed consent and epidemiological questionnaire were obtained from all participants enrolled in the study. The clinical characteristics of the patients analyzed in this study are available in Appendix A. In addition, we organized the data regarding environmental/genetic issues obtained from the analyzed patients in Appendix A.

### 2.3. Sample Genotyping

DNA extraction was performed by the phenol-chloroform method in tissue samples. The peripheral blood DNA from women with no cancer was extracted by the salting-out method [10].

The selected SNPs (rs3803662, rs4415084, rs4784227, and rs7716600) were evaluated using TaqMan probes assays from Thermo Fisher Scientific (Waltham, MA, USA) (rs3803662—C__25968567_10; rs4415084—C__26083318_10; rs4784227—C___1480456_10; rs7716600—C__26299725_10). The reactions were performed using 1 μL of DNA in a concentration of 20 ng/μL, 1x assay (Thermo Fisher Scientific), and 1x TaqPath ProAmp master mix (Thermo Fisher Scientific) in a StepOne Plus qPCR System (Thermo Fisher Scientific). The software was set up for Genotyping assay, which adjusts the cycle and allows the allele interpretations.

### 2.4. lncRNA-SNP Regulatory Aspects Prediction

To investigate more about the selected lncRNA-SNPs, we used the database lncRNASNP2 (http://bioinfo.life.hust.edu.cn/lncRNASNP2, accessed on 15 January 2023) to verify the impact of the genomic alteration on miRNA site gain/loss.

Secondary structure prediction of lncRNA-SNP alleles was retrieved using the RNAfold database (http://rna.tbi.univie.ac.at//cgi-bin/RNAWebSuite/RNAfold.cgi, accessed on 15 January 2023) using as parameter minimum free energy (MFE).

### 2.5. Statistical Analysis

For the genotypic frequency tests of the control and patient groups, we used the test of deviations in the proportions of the Hardy–Weinberg theorem by Chi-square. Additionally, we used the odds ratio (OR) calculation and the Chi-square test to assess whether the variables (breast cancer and SNPs) were independent. We also performed a multivariate analysis considering BC prognostic markers, such as age, estrogen/progesterone status, tumor grade, and presence/absence of lymph node metastasis.

Statistical analyses were performed with R software with the *Nortest* and *readxl* packages [11,12]. For all tests described above, *p*-values < 0.05 were considered significant.

## 3. Results

### 3.1. lncRNA-SNP Selection

In order to highlight lncRNAs that are DE and have an SNP associated with BC risk, we performed a two-step analysis. As the first step, we performed a differential expression analysis using RNA-Seq data from the TCGA breast cohort, comparing the sequences from cancer patients and the control group. Among all the lncRNAs, we found 1334 lncRNAs that were DE based on the filter criteria of fold change |>1.5| and *p* < 0.001 (Appendix A).

Based on this, we compared DE lncRNAs and results from previously published GWAS [5] which evaluated 137,045 BC samples and 119,079 controls and found 38,134 BC risk SNPs. As a filter criterion, we also applied the selection of SNPs with minor allele frequency (MAF) > 0.20 to avoid rare alleles. This analysis resulted in 23 lncRNA-SNPs (Appendix A).

Since BC prognosis and treatment vary substantially across different BC subtypes, we carried out a final filter in which, from the **23 lncRNA-SNPs**, we searched for ones that were DE among different BC subtypes. To accomplish this, we compared patients using TCGA RNA-Seq data based on molecular classifications, namely: luminal A, luminal B, HER2-enriched, and basal-like. As a result, we highlighted **14 lncRNA-SNPs** that exhibited the three major filtering criteria (1) DE between tumor and normal samples; (2) SNPs associated with BC risk with MAF > 0.20; (3) DE between BC molecular subtypes (Table 1).

### 3.2. Selected lncRNA-SNPs

Based on the described methodology, among the 14 lncRNA-SNPs, we selected four lncRNA-SNPs with the highest OR values to conduct a case-control study in a Brazilian cohort (Table 1). It is important to highlight that none of the lncRNA-SNPs mentioned here have been previously evaluated in a Brazilian cohort.

The lncRNA-SNP rs7716600 exhibited the highest OR value (1.24), located at the lncRNA named AC093297.2. Based on data in the literature, this SNP had already been associated with BC in different studies [13,14,15]. In our analysis, this lncRNA was upregulated in tumor samples compared to the control group. It is also downregulated in estrogen-positive subtypes and upregulated in HER2-positive subtypes.

Another selected lncRNA, LINC02224, was upregulated in tumor samples compared to the control group and is downregulated in estrogen-positive subtypes. The SNP rs4415084, significantly associated with a GWAS cohort, has already been associated with BC risk in different populations [16,17] but has never been studied in a Brazilian cohort.

Two SNPs (rs4784227/rs3803662) located in the lncRNA cancer susceptibility 16 (CASC16) have been investigated in BC in different populations [18,19,20,21], indicating their possible association with BC risk. Based on our expression analysis, CASC16 is upregulated in tumor samples (logFC = 1.92) and is specifically upregulated in estrogen-positive subtypes. The SNPs exhibit linkage disequilibrium in CEU (Utah Residents from North and West Europe) (D’ = 0.97 and R^2^ = 0.91), making them interesting candidates for exploration in haplotypes.

### 3.3. Hardy–Weinberg Equilibrium and Allele Frequency Comparison

Both case and control samples were in Hardy–Weinberg Equilibrium (HWE) (*p*-value < 0.05) for all the evaluated lncRNA-SNPs in our Brazilian cohort.

Gathering the allele frequency data together, we compared the observed frequency for the minor allele for the four lncRNA-SNPs of our cohort relative to the control group and the estimates of the 1000 Genomes Database for European and African populations. We organized these data in Table 2 below.

### 3.4. lncRNA-SNPs and BC Susceptibility

In order to evaluate the impact of the lncRNA-SNPs and BC susceptibility, we performed a case-control association study based on odds ratio (OR) calculation (Table 3, Table 4, Table 5 and Table 6). Considering rs4415084, we found a significant association value considering the dominant model (Table 4), and for rs7716600 a significant case-control association was observed in the recessive model (Table 6). Both of the lncRNA-SNPs-associated genotypes were related to high BC risk.

For rs3803662 (Table 3) and rs4784227 (Table 5), when analyzed individually, we did not find significant results testing the genotypes for codominant, dominant, recessive, and over-dominant models. It is important to highlight that we performed haplotype analysis (Table 7) for rs3803662/rs4784227 since these polymorphisms exhibit a linkage disequilibrium (LD), with rs3803662 considered as the Tag SNP. Using the LDlink database from NCBI (https://ldlink.nci.nih.gov/, accessed on 12 November 2022), the R^2^ estimated for these two lncRNA-SNPs is 0.91. Regarding this analysis, we found that GT haplotype, which is the rare one, exhibited a significant association *p*-value.

### 3.5. lncRNA-SNPs and BC Clinical Parameters

After evaluating the association of lncRNA-SNPs in a case-control study, we divided the BC case samples according to some important clinical parameters in prognosis and treatment: age at diagnosis, estrogen/progesterone receptors status, tumor grade, and lymph node metastasis.

To evaluate age, we classified the samples into two groups: (1) higher than the mean age of diagnosis (57.90 ± 14.51) and (2) lower than the mean age of diagnosis. Estrogen, progesterone, and lymph node metastasis were stratified into two groups (1) positive and (2) negative, and tumor grade was divided into (1) grade I, (2) grade II, and (3) grade III. We found a significant association between rs4415084 and progesterone receptor status (Table 8). Given the recessive and over-dominant models of the genotypes, the presence of the C allele (CC/CT) is associated with the positivity of the progesterone receptor. This could be considered a good prognosis marker since therapy for hormone-positive BC cases is available.

We also found a clinical correlation between rs7716600 and lymph node status (Table 9). In the dominant model, the presence of allele A is related to the absence of lymph node infiltration. This is also a good prognosis marker.

### 3.6. lncRNA TCGA Expression Data

We evaluated the expression data of the selected lncRNAs LINC02224, CASC16, and AC093297.2 using TCGA cohort data presented in the TANRIC database (https://ibl.mdanderson.org/tanric/_design/basic/main.html, accessed on 12 November 2022). We focused our analysis on the same clinical parameters evaluated in the association study.

LINC02224 is downregulated in the progesterone-negative subtype (Figure 1), and based on our results, the presence of the C allele (CC/CT genotypes) is associated with progesterone status positivity.

### 3.7. lncRNA-SNPs Regulatory Effects

Using lncRNASNP2, we looked for potential miRNA site gain/loss in the analyzed lncRNA-SNPs. This information is important considering the regulatory roles of miRNAs and lncRNAs in ceRNA networks. We only found data for rs3803662 in the database, which exhibited a miRNA site gain for hsa-miR-196a-3p and miRNA site loss for hsa-miR-4524a-3p. Based on expression data of the TCGA BRCA cohort, miR-196a-3p is upregulated in tumor samples compared to the control group; moreover, miR-4524a-3p is shown to be upregulated in tumor samples with high grades compared to low-grade ones.

Another important aspect considering lncRNA function is based on its secondary structure. This can directly impact target interaction and can result in the gain/loss of target sites. With this in mind, using the RNAfold prediction database and the lncRNA-SNPs sequences, we investigated whether these genomic alterations could lead to structural variations. For this analysis, we used the parameter minimum free energy (MFE), and the result can be observed in Figure 2 below.

This information can be relevant considering the RNA partners these lncRNAs might have in a cellular context. For example, the lncRNA CASC16 is predicted to have several RNA Binding Protein (RBP) domains for TATA-Box Binding Protein Associated Factor 15 (TAF15), FUS, and UPF1. These proteins have essential regulatory roles and are also related to the BC context. Thus, disrupting the wild-type secondary structure can impact these protein bindings, deregulating some cellular processes.

## 4. Discussion

In recent years, lncRNAs have gained increased attention due to their versatile regulatory roles and use as biomarkers. These molecules have been implicated in several diseases, including BC [22]. In order to elucidate the role of lncRNA in the BC context, experiments are being conducted to evaluate lncRNA expression, epigenetic profiles, and genomic alterations, including SNPs.

A previous study from our group found a significant association of rs527616 (C > G), located in the lncRNA AQP4-AS1, with BC in a Brazilian population. We found that CG heterozygotes were above the expected, and the over-dominant model is the best one to explain our results (OR: 1.70, IC 95%: 1.23–2.34, *p* < 0.001). Furthermore, the SNP was associated with age at BC diagnosis, and the risk genotype was more frequent in the older age group [23]. Building on these results, we conducted a comprehensive integrative study to find lncRNA-SNPs associated with BC.

Taking together expression and genomic data, we selected lncRNA-SNPs that also exhibited significant differential expression between tumor and non-tumor tissue in the TCGA BRCA cohort and looked for ones that had been previously associated with BC susceptibility using the GWAS database. We chose the four lncRNA-SNPs with the highest OR values in the GWAS study for evaluation in a Brazilian BC population.

We observed that the lncRNA-SNPs rs4415084 and rs7716600 exhibited a significant association *p*-value in the dominant and recessive models, respectively (Table 4 and Table 6). Both are related to an increased risk of BC development in the analyzed population. The rs4415084 has already been highlighted as relevant in the development of BC in Caucasian Slavic [24], Chinese [25,26], and European populations [27]. Additionally, lncRNA-SNP rs4415084 was significantly associated with progesterone receptor status considering the recessive and over-dominant models (Table 8), the presence of the C allele (CC/CT) being associated with the positivity of the progesterone receptor. This may be a good prognosis marker since hormone-based therapy is available.

The rs7716600 is less explored in the literature. In our study, this lncRNA-SNP was associated with high BC risk (Table 6), also lymph node negative status (Table 9). Kim and colleagues [28] studied a Korean population and found that rs7716600 was significantly associated with breast cancer risk, and Quigley et al. [13] found an association with this lncRNA-SNP in estrogen-positive tumors.

The lncRNA-SNPs rs4784227/rs3803662 are in strong LD and were evaluated in haplotypes. We found a significant result for the haplotype GT in case-control study. These lncRNA-SNPs have already been investigated in the BC context. In a Chinese population, rs3803662 was associated with a protective role in breast cancer risk, while rs4784227 increased breast cancer susceptibility at age > 50 years [29].

Looking for lncRNA-SNPs rs4784227/rs3803662 and secondary structure changes, we found that rs4784227 alleles exhibited significantly distinct structures (Figure 1). This alteration can lead to the gain/loss of target interaction sites. CASC16, which contains both rs4784227 and rs3803662, seems to have several RBP sites. According to CLIP data, CASC16 interacts strongly with TAF15 [30], UPF1 [31], and FUS [32]. TAF15 plays a role in RNA polymerase II gene transcription as a component of a distinct subset of multi-subunit transcription initiation factor TFIID complexes [33]. UPF1 is a protein that is part of a post-splicing multiprotein complex involved in both mRNA nuclear export and mRNA surveillance, and data evaluating UPF1 and other lncRNA interactions provide a fundamental basis for cell transformation and tumorigenic growth [34]. Finally, FUS is a multifunctional protein component of the heterogeneous nuclear ribonucleoprotein (hnRNP) complex, which is associated with triple-negative BC progression [35].

Looking deeper into the potential regulatory role of these alleles in the BC context, we found that rs3803662 exhibited a miRNA site gain for hsa-miR-196a-3p and miRNA site loss for hsa-miR-4524a-3p. miR-196a-3p is considered an estrogen-regulated miRNA and is a robust prognostic factor for patients with advanced and post-menopausal ER+ disease [36,37]. Molecular aspects of miR-4524a-3p have never been investigated in BC; however, based on expression data, this miRNA is upregulated in high-grade tumor samples compared to low-grade ones. Nevertheless, the case-control study failed to find a correlation in the Brazilian population.

Gathering all these data together, we can emphasize that the strategy of integrative analysis (differential expression and GWAS) in mapped regions of lncRNAs is a good strategy to suggest candidates for deeper analysis. These results suggest that lncRNA-SNPs have potential utility as prognostic markers of BC and must be investigated in greater detail, focusing on their molecular role in the cancer context.

## 5. Conclusions

SNP-type genomic changes in lncRNAs play an important role in disease development. In this work, based on an integrated bioinformatics methodology, we highlight four lncRNAs-SNPs with a potential role in breast carcinogenesis. We identified the high risk of developing the disease by analyzing the rs4415084 and rs7716600 in case-control study. These same SNPs also showed an association with important prognostic parameters of the disease. When analyzing the haplotypic region containing the SNPs of lncRNA CASC16, it was also possible to observe a significant association.

This study is the first to demonstrate the role of these lncRNA-SNPs in the Brazilian population, emphasizing the need to expand research in the area.

## Figures and Tables

**Figure 1 genes-14-00971-f001:**
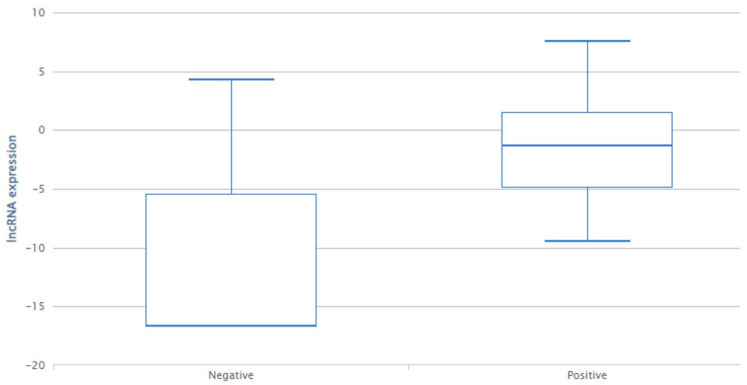
LncRNAs expression from TCGA BRCA cohort. LINC02224 is downregulated in the progesterone-negative subtype (*p* < 0.001).

**Figure 2 genes-14-00971-f002:**
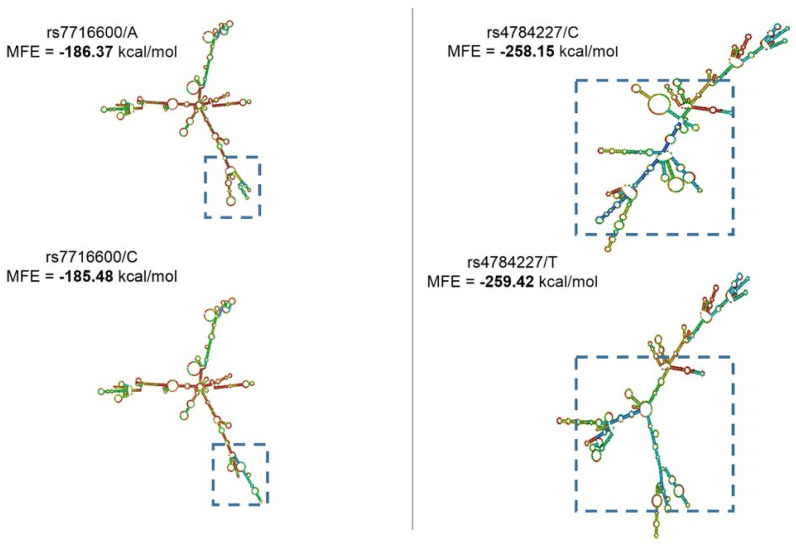
Secondary structures of lncRNA-SNPs that exhibited differences in minimum free energy (MFE). It is highlighted in the figure the fraction with structural change in the lncRNA molecule. In rs7716600, it is possible to note that the presence of C allele results in a decrease in MFE. In contrast, it is observed that the presence of the T allele in the SNP rs4784227 is associated with an increase in MFE.

**Table 1 genes-14-00971-t001:** Information about the 14 selected lncRNAs-SNPs.

LncRNA	logFC	*p*-Value ^1^	DE Subtype	SNP	OR	*p*-Value ^2^	C.I	MAF
AC020916.1	1.99	1.27 × 10^−147^	<ER+	rs2594714	1.04	1 × 10^−8^	1.01–1.05	0.42
AC093297.2	1.86	2.70 × 10^−15^	<ER+/>HER+	rs7716600	1.24	7 × 10^−7^	1.14–1.34	0.27
AL358075.2	1.45	3.39 × 10^−11^	<HER2+	rs1707302	1.04	3 × 10^−8^	1.02–1.05	0.37
AQP4-AS1	0.87	0.000000000937	>ER+	rs527616	1.03	7 × 10^−15^	1.02–1.05	0.23
CASC16	1.92	0.00000000000000957	>ER+	rs4784227/rs3803662	1.23	4 × 10^−117^	1.2–1.25	0.25
CRYZL2P-SEC16B−201	−0.80	1.90 × 10^−12^	>HER2+	rs575908	1.03	3 × 10^−6^	0.019–0.045	0.35
LINC-PINT	−0.69	0.00000114	>HER−	rs68056147	1.05	5 × 10^−13^	0.037–0.064	0.24
LINC02224	3.92	3.14 × 10^−100^	<ER+/>HER+	rs4415084	1.17	8 × 10^−11^	1.11–1.22	0.46
LINC01977	1.48	5.16 × 10^−19^	<ER+	rs745570	1.03	4 × 10^−10^	1.01–1.05	0.38
LINC00511	1.72	2.70 × 10^−14^	>ER+	rs11652463	1.04	8 × 10^−8^	0.025–0.055	0.43
LINC00536	1.05	0.0000143	>ER+	rs13267382	1.03	2 × 10^−11^	1.03–1.07	0.34
LINC00578	1.91	2.70 × 10^−16^	<ER+/<HER+	rs7430456	1.02	5 × 10^−6^	0.017–0.042	0.44
MEG3	−1.93	2.53 × 10^−47^	>ER+	rs2295389	1.03	2 × 10^−6^	0.02–0.049	0.27
MIR4435-2HG	0.88	6.77 × 10^−10^	<HER2+	rs200484318	1.04	7 × 10^−6^	0.024–0.062	0.36

Legend: *p*-value ^1^—*p* value referent to differential expression analysis considering BC tumor × Normal breast/DE Subtype—BC subtype that showed significant different of the lncRNA expression/OR—Odds-Ratio/*p*-value ^2^—*p* value referent to GWAS analysis/C.I.—Confidence interval/MAF = Minor allele frequency. SNPs colored in red are the selected ones.

**Table 2 genes-14-00971-t002:** Altered allele frequencies in Brazilian cohort data and in 1000 Genomes Study Europe (EUR) and Africa (AFR).

lncRNA-SNP	Alt Allele Brazilian Cohort	Alt Allele 1000 Genomes Europe	Alt Allele 1000 Genomes African
rs7716600	0.77	0.79	0.83
rs4784227	0.26	0.25	0.04
rs3803662	0.68	0.70	0.43
rs4415084	0.44	0.40	0.65

Altered alleles frequencies in 1000 Genomes projects.

**Table 3 genes-14-00971-t003:** Case-control results of rs3803662 in the analyzed models.

rs3803662	Case (*n* = 273)	Control (*n* = 367)	OR (95%)	*p*-Value
GG	125	172	1.00	0.6
AG	121	151	0.91 (0.65–1.26)	
AA	27	44	1.18 (0.70–2.02)	
Models				
Dominant				
GG	125	172	1.00	0.79
AG/AA	148	195	0.96 (0.70–1.31)	
Recessive				
AA	27	44	1.24 (0.75–2.06)	0.4
GG/AG	246	323	1.00	
Overdominant				
AG	121	151	0.88 (0.64–1.21)	0.42
GG/AA	152	216	1.00	

**Table 4 genes-14-00971-t004:** Case-control results of rs4415084 in the analyzed models.

rs4415084	Case (*n* = 285)	Control (*n* = 357)	OR (95%)	*p*-Value
CC	104	100	1.00	
CT	132	182	1.43 (1.01–2.04)	0.065
TT	49	75	1.59 (1.01–2.50)	
Models				
Dominant				
CC	104	100	1.00	0.022
CT/TT	181	257	1.48 (1.06–2.06)	
Recessive				
TT	49	75	1.00	0.22
CC/CT	236	282	1.28 (0.86–1.91)	
Overdominant				
CT	132	182	1.00	0.24
CC/TT	153	175	1.21 (0.88–1.65)	

**Table 5 genes-14-00971-t005:** Case-control results of rs4784227 in the analyzed models.

rs4784227	Case (*n* = 270)	Control (*n* = 355)	OR (95%)	*p*-Value
CC	153	199	1.00	0.81
CT	94	130	1.06 (0.76–1.49)	
TT	23	26	0.87 (0.48–1.58)	
Models				
Dominant				
CC	153	199	1.00	0.88
CT/TT	117	156	1.03 (0.75–1.41)	
Recessive				
TT	23	26	0.85 (0.47–1.52)	0.58
CC/CT	247	329	1.00	
Overdominant				
CT	94	130	1.08 (0.78–1.51)	0.64
CC/TT	176	225	1.00	

**Table 6 genes-14-00971-t006:** Case-control results of rs7716600 in the analyzed models.

rs7716600	Case (*n* = 254)	Control (*n* = 338)	OR (95%)	*p*-Value
CC	153	196	1.00	0.052
AC	94	118	0.98 (0.69–1.38)	
AA	7	24	2.68 (1.12–6.38)	
Models				
Dominant				
CC	153	196	1.00	0.58
AC/AA	101	142	1.10 (0.79–1.53)	
Recessive				
AA	7	24	1.00	0.015
CC/AC	247	314	2.70 (1.14–6.36)	
Overdominant				
AC	94	118	0.91 (0.65–1.28)	0.6
CC/AA	160	220	1.00	

**Table 7 genes-14-00971-t007:** Haplotype results obtained from rs44150082/rs4784227 linkage region.

	rs3803662	rs4784227	Frequency	OR (95%)	*p*-Value
Haplotype 1	G	C	0.67	1.00	
Haplotype 2	A	T	0.25	1.03 (0.80–1.33)	0.81
Haplotype 3	A	C	0.0735	0.99 (0.64–1.52)	0.96
Haplotype 4	G	T	0.0081	0.20 (0.04–0.99)	0.049

**Table 8 genes-14-00971-t008:** Progesterone status association for rs4415084 in the analyzed models.

rs4415084	PR Negative (*n* = 62)	PR Positive (*n* = 166)	OR (95%)	*p*-Value
CC	19	56	1.0	0.062
CT	36	72	0.68 (0.35–1.31)	
TT	7	38	1.84 (0.71–4.81)	
Models				
Dominant				
CC	19	56	1.00	0.66
CT/TT	43	110	0.87 (0.46–1.63)	
Recessive				
TT	7	38	2.33 (0.98–5.54)	0.041
CC/CT	55	128	1.00	
Overdominant				
CT	36	72	0.55 (0.31–1.00)	0.048
CC/TT	26	94	1.00	

**Table 9 genes-14-00971-t009:** Lymph node status association for rs7716600 in the analyzed models.

rs7716600	Lymph Node Status NEG (*n* = 125)	Lymph Node Status POS (*n* = 90)	OR (95%)	*p*-Value
CC	68	61	1.00	0.14
AC	53	27	0.57 (0.32–1.01)	
AA	4	2	0.56 (0.10–3.15)	
Models				
Dominant				
CC	68	61	1.00	0.047
AC/AA	57	29	0.57(0.32–1.00)	
Recessive				
CC/AC	121	88	1.00	0.66
AA	4	2	0.69 (0.12–3.84)	
Overdominant				
CC/AA	72	63	1.00	0.062
AC	53	27	0.58 (0.33–1.03)	

## Data Availability

Data available upon request.

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
