# Peer review of "LncRNA-SNPs in a Brazilian Breast Cancer Cohort: A Case-Control Study"

_genes, 2023, doi:10.3390/genes14050971_

Round 1
Reviewer 1 Report
1. The classification of breast cancer is generally divided into invasive cancer, non-invasive cancer and other rare breast cancer, while invasive cancer is divided into invasive non-specific cancer and invasive specific cancer. In this study, whether the 306 breast cancer patients included should be stratified according to the classification of cancer, please make a brief explanation.
2.In this study, the reasons for choosing the hospital the Hospital Nossa Senhora das Graças are not stated. Are the hospitals of the study subjects representative of the whole of Brazil? If so, please provide additional information. If not, please explain the reasons for choosing this hospital.
3. In the results of this study, there is no general demographic information table of the subjects, and the maximum and minimum age of the subjects are supplemented; Whether there is menopause; Breast cancer related diseases, such as taking hormone drugs, family genetic factors; Living habits, such as smoking, drinking, high-fat diet, sedentary and so on.
4. In the control group of this study, the article only selected peripheral blood samples of 312 women without cancer history, whether the 312 control group can represent a random sample of breast cancer risk groups in Brazil, please briefly explain.
5. In the results and conclusions of this study, we only described the association of microRNA, lncRNA and SNP sites, but did not verify the process, whether it is necessary to increase the relevant experiments or bioinformatics analysis to enhance its credibility.
Author Response
- The classification of breast cancer is generally divided into invasive cancer, non-invasive cancer and other rare breast cancer, while invasive cancer is divided into invasive non-specific cancer and invasive specific cancer. In this study, whether the 306 breast cancer patients included should be stratified according to the classification of cancer, please make a brief explanation.
A: The reviewer's comment is quite pertinent and, indeed, of a direct impact on determining the patient's prognosis. In this way, we carried out a thorough analysis of the clinical data of the patients analyzed, and we verified that some diagnoses were inconclusive. Therefore, we opted to remove them from the statistical analyses. Initially, we used 306 patient samples, and after the new filtering, 291 samples remained. This information was corrected in the manuscript ("Study Cohort" section). Therefore, we performed the case-control association analysis again for all lncRNAs-SNPs studied. The new data is highlighted in the manuscript in yellow. Additionally, we have organized clinical data obtained from each patient in Supplementary Table 2. In this way, we added the following information in the manuscript (Study Cohort section): “The clinical characteristics of the patients analyzed in this study are available in supplementary table 2. In addition, we organized the data regarding environmental/genetic issues obtained from the analyzed patients in supplementary table 3”
- In this study, the reasons for choosing the hospital the Hospital Nossa Senhora das Graças are not stated. Are the hospitals of the study subject’s representative of the whole of Brazil? If so, please provide additional information. If not, please explain the reasons for choosing this hospital.
A: The “Hospital Nossa Senhora das Graças” has been our collaborator for over 30 years. The medical team directly assists us in the collection of biological samples, as well as in obtaining informed consent and clinical information from patients. It is a reference hospital in treating breast neoplasms in the city of Curitiba, as well as the Paraná State and other regions in South Brazil. Patients were compared with healthy donors in the same geographic area, which is not representative of the totality of the Brazilian population. For example, previous studies described around 80% of the population from Curitiba self-reporting as Euro-descendant (IBGE, 2013), which agrees with previous local genetic studies (Probst et al., 2000, Braun-Prado et al., 2010).
Brazil is a continental country with diverse cultures and genetic backgrounds in different regions. Our study brings information about patients treated in a specific region of Brazil. Although genomic information to assess ancestry was unavailable for all individuals, we included the patients’ self-reported records about ancestry. In both groups, patients, and control, almost 85% declared themselves as white and 10% as predominant African ancestry.
Additionally, a subset of patients of our tumor bank was genotyped using SNP chip Illumina Infinium QC Array (Illumina Inc., CA), which contains 15,949 markers (including ~3,000 ancestral informative markers (AIMs). Based on these ancestry data, the patients of this study clustered with the European-defined group from the 1000 Genome Project and with the Admixed Americans main group, mainly composed of Colombians and Mexicans, highlighting the marked ancestral heterogeneity of the population (Sugita et al. Oncotarget, v.10, p.6184-6203, 2019).
- In the results of this study, there is no general demographic information table of the subjects, and the maximum and minimum age of the subjects are supplemented; Whether there is menopause; Breast cancer related diseases, such as taking hormone drugs, family genetic factors; Living habits, such as smoking, drinking, high-fat diet, sedentary and so on.
A: We are grateful for the comment made by the reviewer. In fact, this is very important information for a better characterization of the population. We reviewed the medical records of the patients analyzed in the study and collected the information when present in the clinical records available at the hospital. It was not possible to gather additional information regarding all patients, but we organized all the collected data as Supplementary Table 3. Therefore, we added the following information in the manuscript (Study Cohort section): “The clinical characteristics of the patients analyzed in this study are available in supplementary table 2. In addition, we organized the data regarding environmental/genetic issues obtained from the analyzed patients in supplementary table 3”
- In the control group of this study, the article only selected peripheral blood samples of 312 women without a cancer history, whether the 312 control group can represent a random sample of breast cancer risk groups in Brazil, please briefly explain.
A: Female volunteers were recruited among healthy volunteer bone marrow donors who agreed to be part of the control group and that informed on the absence of personal and familiar cases of cancer. We made this information clearer in the manuscript (in yellow in the submitted file): "As a control group, we used peripheral blood samples of 370 women who were recruited on a voluntary basis and who declared they had no personal and family history of cancer with a mean age of 47.66 ± 4.69 years.
- In the results and conclusions of this study, we only described the association of microRNA, lncRNA and SNP sites, but did not verify the process, whether it is necessary to increase the relevant experiments or bioinformatics analysis to enhance its credibility.
A: Thanks to the reviewer for the comment. We included this topic in our article with the aim of raising hypotheses about the impact that SNPs on lncRNAs could result in the regulation of gene expression. The hypotheses we discussed can be experimentally validated in future works.

Reviewer 2 Report
Introduction
- The introduction could benefit from proof reading by a native/fluent English speaker. In places, the English grammar makes it difficult to fully comprehend the point.
Methods
- Section 2.1: This section talks about performing differential expression using RNASeq data but does not clearly define what was selected using this process (mRNA transcripts/ lncRNAs). One assumes that the RNASeq data was used to select differentially expressed lncRNAs, but this is not clear until the second paragraph.
Results
- Figure 1 could be replaced with a higher resolution version of the figure.
- Figure legends could be more descriptive to better describe the figure, particularly figure 2.
Discussion
- For clarity, gene names could be described in full before being abbreviated.
- Line 290 the ‘a’ in “and also” is a different font/size.
Line 294: Brazilian should have a capital “B”
Overall, the introduction and discussion, particularly, could benefit from additional grammatical editing. The methods and results sections were clearer.
Author Response
Introduction
- The introduction could benefit from proof reading by a native/fluent English speaker. In places, the English grammar makes it difficult to fully comprehend the point.
A: We appreciate the reviewer's comment, and to improve the quality of the work, we sent the entire manuscript for English grammar review.
Methods
- Section 2.1: This section talks about performing differential expression using RNASeq data but does not clearly define what was selected using this process (mRNA transcripts/ lncRNAs). One assumes that the RNASeq data was used to select differentially expressed lncRNAs, but this is not clear until the second paragraph.
A: In this work, we used the total transcriptomic expression matrix of the TCGA patients, and from the differentially expressed data, we identified which were lncRNAs. We agree that the information was missing in our manuscript, and we have added information regarding this methodology (in yellow in the manuscript). The added text:"As the first step, we performed a differential expression analysis using RNA-Seq HTSeq-FPKM from the TCGA Breast Cohort using as cut-offs fold change |>1.5| and p <0.001, and then we identified the ones that were classified as lncRNA", integrates section 2.1 of the methods.
Results
- Figure 1 could be replaced with a higher-resolution version of the figure.
A: We submitted a new version of Figure 1 with 600 dpi.
- Figure legends could be more descriptive to describe the figure, particularly figure 2 better.
A: We agree with the reviewer that the subtitle could be improved. Therefore, in the file submitted as "figure legends", modifications were made to the legends.
Discussion
- For clarity, gene names could be described in full before being abbreviated.
A: We revised the mRNAs cited in this work and added the name in full in its first citation. Modifications are in bold in the submitted manuscript. Added the name "TATA-Box Binding Protein Associated Factor 15" to the TAF15 gene. Among the cited lncRNAs, only CASC16 has already fully established its name. This information was also added to the manuscript. "Cancer susceptibility 16" is the added information.
- Line 290 the ‘a’ in “and also” is a different font/size.
A: Addressed.
- Line 294: Brazilian should have a capital “B”
A: Addressed
- Overall, the introduction and discussion, particularly, could benefit from additional grammatical editing. The methods and results sections were clearer.
A: We agree with the reviewer that the grammatical part needed some adjustments, and in that sense, the manuscript was sent for grammatical review of the English language.
